**Citation:** *Molecular Systems Biology* 9:689
www.molecularsystemsbiology.com

# Protein synthesis rate is the predominant regulator of protein expression during differentiation

**Anders R Kristensen, Joerg Gsponer and Leonard J Foster\***

Department of Biochemistry and Molecular Biology, Centre for High-Throughput Biology, University of British Columbia, Vancouver, British Columbia, Canada
\* Corresponding author. Department of Biochemistry and Molecular Biology, University of British Columbia, 2125 East Mall, Vancouver, British Columbia, Canada V6T 1Z4. Tel.: + 1 604 822 8311; E-mail: foster@chibi.ubc.ca

**External perturbations, by forcing cells to adapt to a new environment, often elicit large-scale changes in gene expression resulting in an altered proteome that improves the cell's fitness in the new conditions. Steady-state levels of a proteome depend on transcription, the levels of transcripts, translation and protein degradation but system-level contribution that each of these processes make to the final protein expression change has yet to be explored. We therefore applied a systems biology approach to characterize the regulation of protein expression during cellular differentiation using quantitative proteomics. As a general rule, it seems that protein expression during cellular differentiation is largely controlled by changes in the relative synthesis rate, whereas the relative degradation rate of the majority of proteins stays constant. In these data, we also observe that the proteins in defined sub-structures of larger protein complexes tend to have highly correlated synthesis and degradation rates but that this does not necessarily extend to the holo-complex. Finally, we provide strong evidence that the generally poor correlation observed between transcript and protein levels can fully be explained once the protein synthesis and degradation rates are taken into account.**
*Molecular Systems Biology* **9:** 689; published 17 September 2013; doi:10.1038/msb.2013.47
*Subject Categories:* proteomics; differentiation & death
*Keywords:* differentiation; macromolecular complexes; protein turnover; proteomics; systems biology

## Introduction

Proteins are not stable constituents in the cell; instead, they are continuously synthesized and degraded, leading to different turnover rates for individual proteins. An analogy of this that represents any given protein is a bathtub with an open drain, where the amount of water in the tub corresponds to the amount of protein, the water coming in from the faucet represents the synthesis rate, the water exiting through the drain represents the degradation rate and the change in the water level with time represents the change in expression of the protein (Figure 1A). If the inflow and outflow rate are equal, then the level in the tub stays constant; however, the water is still exchanged (turned over) with a given velocity and similarly, proteins will also have different turnover rates. In order to become more fit for a new state brought on by external perturbation, the cell needs to change the expression levels of many proteins through the regulation of a number of cellular processes, including transcription, protein synthesis and protein degradation. The protein synthesis rate has been shown to be regulated by microRNAs (Selbach *et al*, 2008), mRNA change and different mRNA sequence features, whereas the protein degradation rate is predominantly regulated by the ubiquitin-proteasomal system (King *et al*, 1996; Larance *et al*, 2013). Proteins with fast turnover rates

are generally characterized by having low abundance (Schwanhäusser *et al*, 2011; Boisvert *et al*, 2012), being intrinsically unstructured (Prakash *et al*, 2004; Gsponer *et al*, 2008), aggregation prone (De Baets *et al*, 2011; Gsponer and Babu, 2012) and involved in signal transduction and transcriptional activation (Legewie *et al*, 2008; Yen *et al*, 2008; Boisvert *et al*, 2012).

The last decade has seen the discovery of a number of characteristics defining the control of absolute expression level in bacteria, mouse and human under steady-state conditions, revealing how the absolute expression level is mainly controlled by the protein synthesis rate, with the degradation rate having only a minimal contribution (Lu *et al*, 2006; Brockmann *et al*, 2007; Vogel *et al*, 2010; Maier *et al*, 2011; Schwanhäusser *et al*, 2011). Much less information is available regarding the contribution of these processes to the regulation of protein expression change when the proteome needs to be rearranged, such as in response to external perturbation. Interestingly, studies that have examined the contribution of mRNA changes to changes in protein abundance have generally found a relatively poor correlation between them (de Godoy *et al*, 2008; Fournier *et al*, 2010; Lee *et al*, 2011; Maier *et al*, 2011; Munoz *et al*, 2011).

Protein metabolism has traditionally been investigated using isotope-labeled amino acids (Schoenheimer *et al*,

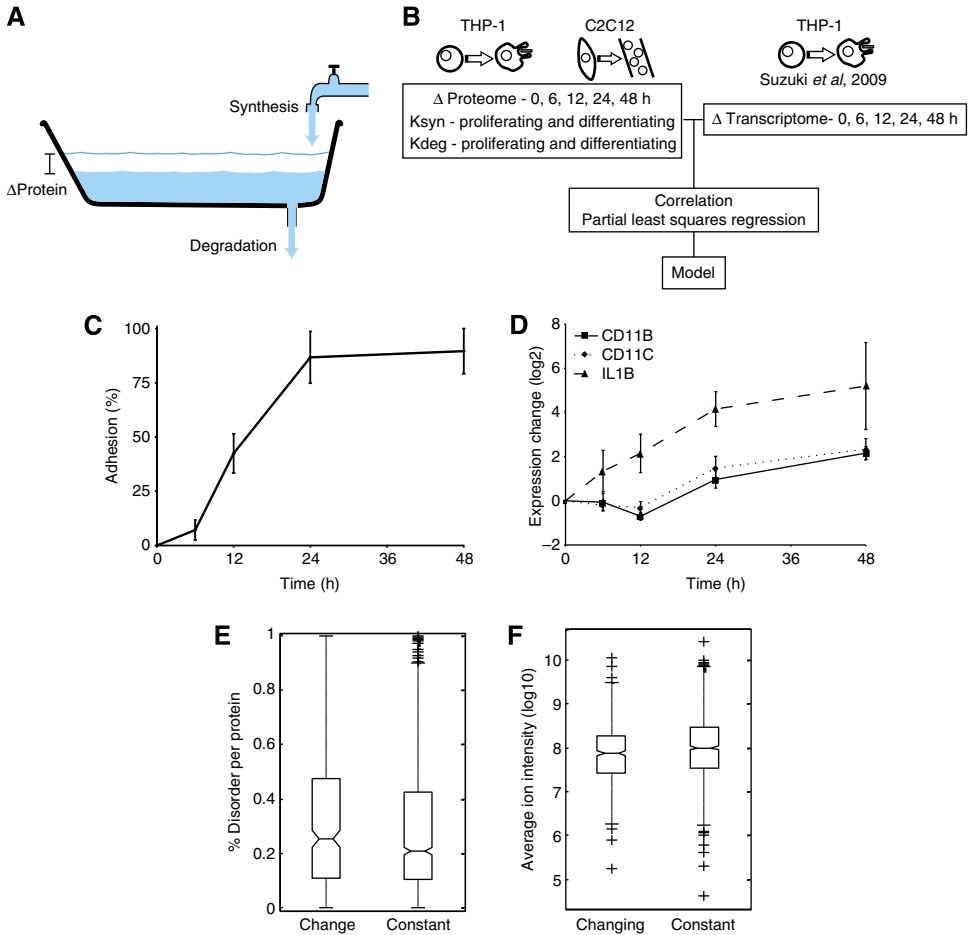

**Figure 1** Proteome changes reflect the phenotype. (**A**) Schematic drawing of the regulation of protein expression change. Water flow rate from faucet resembles protein synthesis rate, draining rate resembles protein degradation rate and water level change resembles protein expression change. (**B**) Changes in overall protein expression, protein synthesis and protein degradation rates were measured during proliferation and differentiation of THP-1 and C2C12 cells. These data were analyzed on their own and also in combination with available transcriptomic data for THP-1 differentiation (Suzuki *et al*, 2009). (**C**) Fraction of THP-1 cells adhered versus time after differentiation was initiated by the addition of 25 nM PMA. (**D**) Relative expression changes of differentiation markers during THP-1 differentiation, as measured by SILAC in the proteomic data. Error bars in (**C**, **D**) indicate standard deviation of three biological replicates. (**E**) Proteins changing expression during differentiation are significantly ($P = 0.03$, Wilcoxon–Mann–Whitney test) enriched in intrinsically disordered proteins. (**F**) Proteins changing expression following perturbation are significantly ($P = 2.8 \times 10^{-5}$, Wilcoxon–Mann–Whitney test) enriched in low abundant proteins. For the boxplots in (**E**, **F**), the line denotes the median, the box denotes the first and third quartiles, the notch denotes the 5% significance level and the whiskers extend to the most extreme values, excluding outliers.

1939); for radionuclides, these were typically detected by autoradiography. Recently, it has become possible to measure the synthesis rate and/or the degradation rate of individual proteins through the use of stable isotope-labeled amino acids and mass spectrometry (Pratt *et al*, 2002; Selbach *et al*, 2008; Cambridge *et al*, 2011; Schwanhäusser *et al*, 2011; Boisvert *et al*, 2012). These experiments are often very time intensive, since they require incorporating the isotope-labeled amino acids for varied length of time followed by quantifying the amount of incorporation. At the end, the synthesis rate can be calculated as a product of the incorporation of labeled amino acids over time. These types of experiments have provided numerous new biological insights, for instance that a single microRNA can repress the translation rate of hundreds of proteins (Selbach *et al*, 2008), that the degradation rate of proteins is conserved between organisms (Cambridge *et al*, 2011) and that the degradation rate can vary for a protein according to the cellular compartment in which it is present

(Lam *et al*, 2007; Boisvert *et al*, 2012). Such efforts, however, have only focused on one aspect of the system, considering neither how synthesis and degradation may be related nor how these parameters are changed in response to stimuli.

Cellular differentiation is accomplished through a precisely orchestrated process where a proliferating cell stops dividing and acquires a new phenotype changing the proteins that it is expressing. The human THP-1 myelomonocytic leukemia cell line has widely been used as a model system to characterize the process of differentiation, since stimulation with phorbol 12-myristate 13-acetate (PMA) induces differentiation into a mature macrophage-like phenotype that no longer proliferates and is characterized by its ability to adhere to substrata, release $O_2^-$, and perform phagocytosis (Auwerx, 1991). Similarly, the murine C2C12 myoblast cell line has also widely been used to characterize the differentiation process, since it can be differentiated by depriving confluent cells of serum, resulting in multinucleated myotubes.

Here, our aim has been to characterize the various factors affecting regulation of protein expression change during cellular differentiation by measuring the synthesis and degradation rates of proteins in proliferating and differentiating THP-1 and C2C12 cells using quantitative proteomics. By comparing these parameters with mRNA and protein expression change at multiple time points, we are able to construct a predictive model that highlights the contribution of different cellular processes to the regulation of the expression change of individual proteins during differentiation (Figure 1B).

## Results

### Quantitative proteomics reveals mechanisms behind phenotypic changes

The primary system used here was the human THP-1 myelomonocytic leukemia cell line that can be differentiated from monocytes into macrophage-like cells by stimulation with 25 nM PMA. We used stable isotope labeling with amino acids in cell culture (SILAC) (Ong *et al*, 2002), with two triplex SILAC experiments to cover five time points of differentiation in THP-1 cells, to mass encode the cells. The cells were stimulated with PMA for 0, 6, 12, 24 and 48 h, which causes several morphological changes, including adherence (Figure 1C); biological triplicates were performed for all experiments. Whole-cell lysates from the different conditions were combined as appropriate and the samples were split into two, with the two halves being pre-fractionated, either at the protein level by SDS–PAGE or at the peptide level by isoelectric focusing (IEF), before analysis on an LTQ-OrbitrapXL mass spectrometer. A total of 4977 proteins could be identified from the tandem mass spectra at a false discovery rate (FDR) below 1%. Among the identified proteins were very well-known markers for monocyte to macrophage differentiation such as CD11B, CD11C and IL1B (Figure 1D), demonstrating that the cells were differentiating as expected and that the method is sensitive enough to detect relevant changes.

To identify proteins with significant differential expression, we applied a strict statistical test (ANOVA $P < 0.05$, $S0 = 1$) among the five time points and found 457 proteins with significant changes in expression during differentiation, which could serve as potentially new markers for the differentiation process (see Supplementary Table S1 for complete list). Functional enrichment analysis on the increasing or decreasing proteins revealed the majority of the originally described phenotypic changes associated with differentiation, such as halting of the cell cycle, increased adhesion and increased lysosomal capacity (Table I). In addition, those proteins altered by differentiation tended to be more intrinsically disordered ($P = 0.03$, Wilcoxon–Mann–Whitney test) (Figure 1E) and were generally lower in abundance ($P = 2.8 \times 10^{-5}$, Wilcoxon–Mann–Whitney test) than the unregulated proteins (Figure 1F).

Finally, we decided to complement these data with a completely unrelated system of differentiation so as to allow us to make more general statements about the regulation of protein expression; to this end we performed a similar study using mouse C2C12 cells that were induced to differentiate from myoblasts, by deprivation of serum at confluence, to

**Table I** Selected significantly-enriched cellular processes during monocyte to macrophage differentiation

| Increase | *P*-value | Decrease | *P*-value |
|---|---|---|---|
| Cell membrane | 1.90E − 07 | DNA replication | 6.00E − 11 |
| Cell adhesion | 2.30E − 06 | Cell cycle | 2.70E − 06 |
| Calcium | 2.40E − 05 | Cell division | 5.10E − 03 |
| Secreted | 2.70E − 05 | Secreted | 9.80E − 04 |
| Lysosome | 4.80E − 02 | Zinc finger | 3.30E − 02 |

myotubes for 48 h (Supplementary Table S2). Even though the myoblasts-to-myotubes and monocytes-to-macrophages systems are quite distinct and from different organisms, many of the same mechanisms appear to be employed to achieve the required changes (Supplementary Figure S1).

### Measuring relative synthesis and degradation rates

To gain insight into the factors that regulate protein expression change, we decided to measure the synthesis and degradation rates of proteins in both proliferating and differentiating THP-1 and C2C12 cells, which hereby should allow us to deduce some of the general regulatory mechanisms at work. To increase the throughput of this study, we decided to measure the relative synthesis and degradation rates of the individual proteins by slightly modifying a previously described approach (Boisvert *et al*, 2012). We first completely incorporated two cell populations using light and medium amino acids. Then, by switching the medium population to heavy amino acids at the same time as the induction of differentiation, all newly synthesized proteins are made with heavy forms of amino acids and all proteins being degraded can be monitored as a decrease in medium amino acids, with the light-labeled population acting as an internal control against which the medium and heavy can be compared. These measurements were made in proliferating myoblasts and monocytes, as well as during the differentiation of each cell type, thereby allowing us to deduce the changes in the synthesis and degradation rates of the individual proteins between proliferating and differentiating cells. Biological triplicates of all the above experiments resulted in the identification of 5721 proteins with an FDR of below 1% (Supplementary Table S3). We subsequently calculated the relative degradation and synthesis rates of the individual proteins by Z-transforming the medium/light and heavy/light ratios of the proteins, respectively (see Supplementary materials for details).

With these data in hand, we first asked if the approach of identifying relative degradation and synthesis rates could identify some of the turnover characteristics previously described in studies that had measured absolute protein turnover. We therefore defined unstable proteins as the 20% of proteins with the fastest degradation rate and stable proteins as the 20% of proteins with the slowest degradation rate in both murine and human cells. In this way, we were able to confirm what has been reported previously that unstable proteins are less abundant ($P = 1.7 \times 10^{-30}$, Wilcoxon–Mann–Whitney test), more intrinsic disordered ($P = 1.3 \times 10^{-27}$,

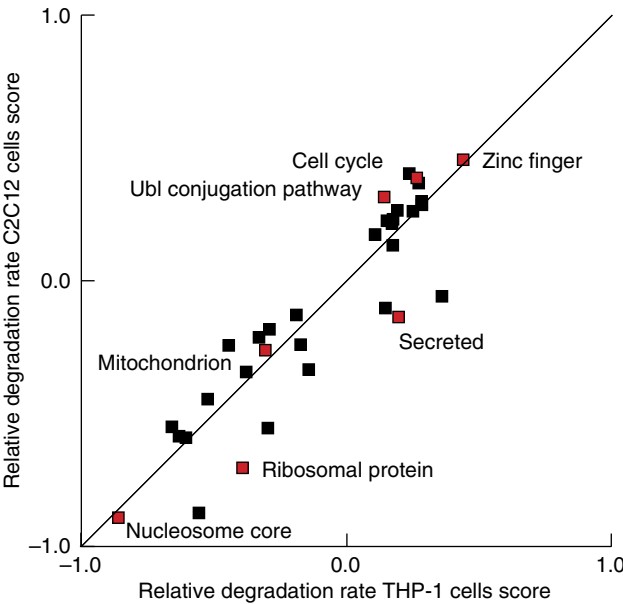

**Figure 2** Relative protein degradation rates capture known protein turnover characteristics. Degradation rates for functional classes of proteins are largely similar ($P < 0.05$) between THP-1 and C2C12, based on the 2D enrichment analysis (Cox and Mann, 2012).

Wilcoxon–Mann–Whitney test) and contain more KEN-box motifs ($P = 0.02$, Fisher's exact test) than stable proteins.

Previous observations have shown how proteins involved in distinct biological processes can display different stabilities (Cambridge *et al*, 2011; Boisvert *et al*, 2012) so, to test this in our own data we performed two-dimensional enrichment analysis between the proliferating C2C12 and THP-1 cells (Cox and Mann, 2012). Briefly, this tests for whether cellular processes are displaying consistent behavior in any of the data dimensions versus the rest of the proteins in the data set. This revealed that protein stability is very consistent regulation between the two cell lines, since the majority of biological processes are located on the diagonal (Figure 2).

## Turnover rates of macromolecular sub-complexes

Having established the veracity of the data from the parallel measurement of synthesis and degradation rates under both proliferating and differentiating conditions in two different cell lines, we next asked whether there were any characteristics common to the proteins that display similar synthesis and degradation rates. As a measurement of similarity of the synthesis and degradation rates of the proteins, we calculated the Euclidian distances between proteins involved in the same biological process in synthesis-degradation space and determined whether this distance was significantly shorter for particular groups of proteins than random chance would dictate. Two proteins with similar synthesis and degradation rates should therefore have short Euclidian distances, whereas protein with very different synthesis and degradation rates will be farther apart. By measuring the distances between proteins involved in the same biological process of both proliferating and differentiating THP-1 and C2C12 cells, it

became especially clear that proteins participating in macromolecular complexes had very similar synthesis and degradation rates (Supplementary Table S4). We could investigate, in a similar way, whether any members of the macromolecular complex displayed different synthesis/degradation from the rest of the members of the complex. Interestingly, we noticed that $30 \pm 1\%$ of the macromolecular complexes had members with different synthesis and/or degradation rates from the other members of the same complex (Supplementary Table S4). An interesting example is RPN10 (aka MCB1 or PSMD4) that displayed different synthesis and degradation from the rest of the proteasome regulatory particle (Figure 3A), whereas all the members of the proteasome core complex displayed similar synthesis and degradation rates.

To investigate the synthesis and degradation rates for proteins that participate in macromolecular complexes in more detail, we decided to directly measure synthesis and degradation within individual complexes using protein correlation profiling-SILAC (PCP-SILAC) and size-exclusion chromatography (SEC) (Kristensen *et al*, 2012). This was accomplished by first completely incorporating two cell populations using light and medium amino acids and subsequently switching the medium population to heavy amino acids and allowing the cells to proliferate for 24 h. However, this time the cells were lysed without the use of detergents and the resulting lysate was separated by SEC into 48 fractions, before the proteins in each individual fraction were digested to peptides and analyzed by LC-MS/MS. Hereby, the individual sub-complexes will be separated out by SEC and elution profiles of the individual proteins can be constructed, using label-free quantitation (Cox and Mann, 2008), from the ion intensities of the light form of each peptide, whereas the relative synthesis and degradation can be determined by the H/L and M/L ratios, respectively (see Supplementary materials for details). By this approach, we were able to identify 2423 proteins in 48 SEC fractions, from which 31% are recorded to participate in macromolecular complexes based on the CORUM database (Supplementary Table S5).

The discovery approach to investigate the synthesis and degradation used here allowed us, for the first time, to unravel whether any relationship exists between protein complex size and the degradation and synthesis rates of the components. While it seems reasonable to expect that the proteins found in larger complexes might be more stable than those in smaller complexes since more energy would be required to replace a large complex than a small one but, surprisingly, there appears to be no correlation between complex size and component protein stability (Supplementary Figure S2).

Next, we validated whether the proteins of the proteasome core subunit also displayed similar synthesis and degradation rates at the sub-complex level. Plotting the elution profile and synthesis and degradation rates of the alpha- and beta-subunits revealed similar synthesis and degradation rates when these co-elute with regulatory subunit, suggesting that the 26S proteasome is degraded as an intact complex (Figure 3B and C). Interestingly, we noticed that around fraction 29 the alpha-subunits displayed an additional peak, which co-eluted with the peaks for the proteasome assembling chaperones (PAC1–PAC4) (Supplementary Figure S3) and is characterized by only containing newly synthesized proteins,

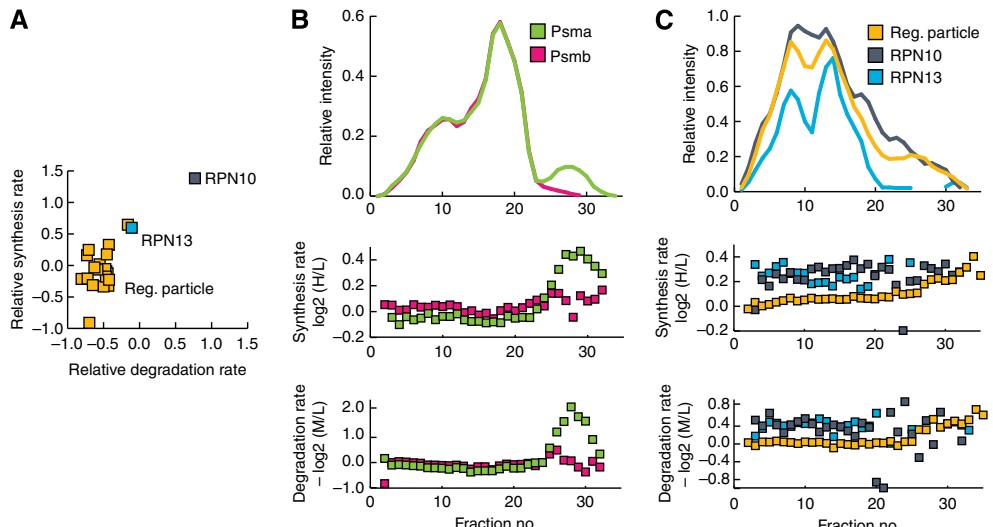

**Figure 3** Synthesis and degradation rates are consistent among members of protein complexes. (**A**) The relative synthesis and degradation rates of the components of the proteasome regulatory particle, which are generally tightly clustered in a scatterplot of synthesis versus degradation rates. (**B**) Top, the median size-exclusion chromatograms of the alpha- and beta-subunits of the proteasome core particle. Middle, the synthesis rate of the alpha- and beta-subunits of the proteasome core particle. Bottom, the degradation rate of the alpha- and beta-subunits of the proteasome core particle. (**C**) Top, the median size-exclusion chromatogram of the regulatory particle of the proteasome, RPN10 and RPN13. Middle, the synthesis rate of regulatory particle of the proteasome, RPN10 and RPN13. Bottom, the relative degradation rates of the regulatory particle of the proteasome, RPN10 and RPN13.

suggesting that we are able for the first time to capture that the proteasome alpha-ring gets *de novo* assembled from only newly synthesized proteins.

In our initial measurements of synthesis and degradation rates, we observed that the ubiquitin receptor RPN10 of the proteasome regulatory particle is synthesized and degraded faster than the rest of the proteins of the regulatory subunit. This could be a result of RPN10 participating in additional complexes that are regulated differently from the proteasome and thus result in distinct average rates for RPN10 or because RPN10 is simply turned over differently from all the other components of the proteasome. To investigate this in more detail, we decided to compare the elution profile and turnover rates of RPN10 with the rest of the proteins within the regulatory particle, revealing that the elution profile of RPN10 is very similar to the rest of the regulatory particle, yet its turnover rate even in that region of the size-exclusion chromatogram was still significantly faster than the rest of the proteins of the regulatory particles $(P = 2.8 \times 10^{-8}$, Wilcoxon–Mann–Whitney test) (Figure 3C). Intriguingly, we noticed that the other ubiquitin receptor of the proteasome RPN13 (Adrm1) displayed similarly high turnover rates while bound to the regulatory particle $(P = 3.7 \times 10^{-8}$, Wilcoxon–Mann–Whitney test), suggesting for the first time that both the ubiquitin receptors RPN10 and RPN13 can exchange with their free forms and that this approach provides a completely novel ability to probe such details.

## Temporal correlation between the transcriptome and the proteome

Many studies have found a poor correlation between changes in mRNA and protein levels in response to perturbation

(de Godoy *et al*, 2008; Fournier *et al*, 2010) but the Central Dogma still suggests that there must be a link. We therefore decided to correlate our data sets of protein expression change and the relative synthesis rate data with a recently published transcriptomic study, which also studied differentiation of THP-1 cells using similar stimuli and time points as our experiments (Suzuki *et al*, 2009). First, we investigated how protein synthesis correlated with the mRNA and protein expression changes, both of which revealed moderate correlation of $0.52 \pm 0.07$ and $0.59 \pm 0.07$, respectively (Figure 4A). The lack of a perfect correlation highlights how protein synthesis is regulated by a number of post-transcriptional processes, and that the levels of proteins within a cell are regulated by processes beyond just protein synthesis.

Few studies have investigated how the correlation between expression change of mRNA and proteins changes over time in response to perturbation, and the correlation has only been investigated after relatively short-term perturbations (Fournier *et al*, 2010; Lee *et al*, 2011). When we examined the correlation of mRNA and proteins expression change at each of the five time points, we observed that the relationship nearly reached a steady state after 24 h differentiation, suggesting a lag between mRNA and protein expression changes and/or that extensive post-transcriptional regulation is taking place during early differentiation (Figure 4B). Interestingly, however, proteins and mRNAs that were being up- or down-regulated during differentiation were much more highly correlated, with < 6% of the significantly changing genes showing anti-correlation at 48 h differentiation, suggesting that post-transcriptional regulation such as miRNA and 3′ and 5′ UTRs is mainly just fine tuning the levels of these proteins after 48 h stimulation. Similarly, 2D enrichment analysis revealed that a generally consistent regulation of the cellular processes in both the transcriptome and the proteome of differentiating THP-1 cells (Figure 4C).

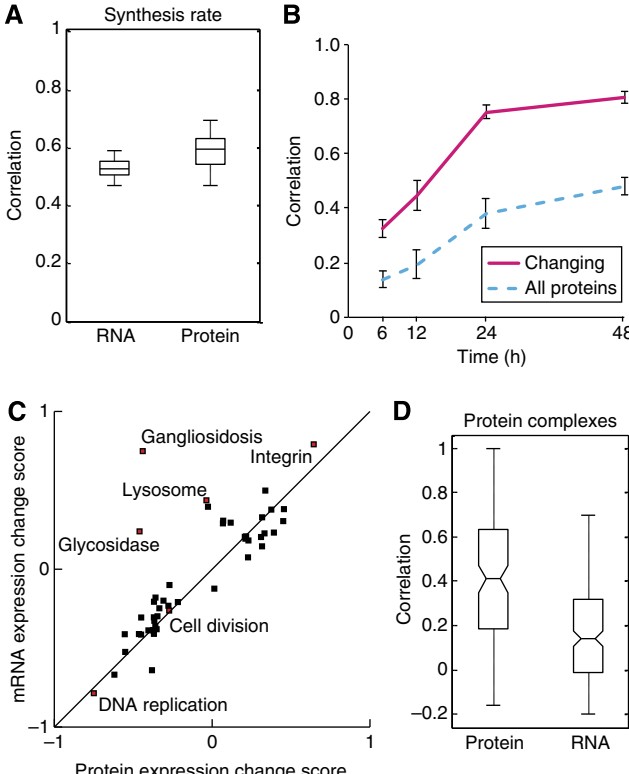

**Figure 4** Correlation between mRNA and protein expression increases over time during monocyte to macrophage differentiation. (**A**) Spread of the correlation between the protein synthesis rate and final changes in mRNA or protein expression changes after 48 h THP-1 differentiation for three biological replicates of each parameter ($n = 9$). The center horizontal line represents the median, the box spans the first through the third quartiles and the whiskers span the most extreme values, excluding outliers. (**B**) A plot of the level of correlation between mRNA and protein expression changes across 48 h of differentiation. The solid pink line represents genes that are changing at the mRNA and protein level whereas the dotted blue line represents all the genes. Error bars denote the standard deviation of three biological replicates of both transcriptomic and proteomic experiments. (**C**) 2D enrichment ($P < 0.05$) of functional classes according to the expression changes observed in the proteome versus the transcriptome. (**D**) Correlation of mRNA versus protein expression changes for members of CORUM complexes during differentiation of THP-1 cells reveals that protein expression change are considerably better predictors for interactions than are mRNA change ($P = 7.9 \times 10^{-14}$, Wilcoxon–Mann–Whitney test) ($n = 125$ and $n = 227$ for protein and mRNA, respectively).

Messenger RNA expression changes have widely been used to predict protein–protein interactions since proteins with similar mRNA expression change are more likely to interact than a random selection of proteins (Jansen *et al*, 2002). Since we have mRNA and protein expression change recorded in the same system at similar time points, we explored what parameters are most predictive for interactivity. We focused only on those proteins annotated as components of complexes in the CORUM database, as this is the most widely accepted gold-standard interaction set. A comparison of how closely the mRNA versus protein expression change tracked for members of CORUM complexes reveals that protein expression changes are vastly more predictive for interactions than are mRNA change ($P = 7.9 \times 10^{-14}$, Wilcoxon–Mann–Whitney test) (Figure 4D). This suggests a novel concept: that if protein expression changes can be measured, then they could be used

to make much more accurate predictions (e.g., of protein–protein interactions) than what mRNA expression changes alone would yield.

## Modeling the control of protein expression

A protein's expression change is regulated by a number of processes, such as RNA transcription, protein synthesis and protein degradation but little information exists about the contribution of each or the combination of any of these processes in the control of protein expression change. If one looks first at the effect of synthesis and degradation rates on overall protein expression, it is obvious that whether the synthesis and degradation rates of a protein are equal, then the expression of the protein is at a steady state, whereas the net protein expression change is the result of the relationship between the synthesis and degradation rates (Figure 5A and B).

We then applied partial least square regression to our own data from differentiating THP-1 cells and similar transcriptional data from Suzuki *et al* (2009) to examine for the first time the contributions of various factors across the board of all proteins to the ultimate changes in protein expression. This revealed that after 48 h of THP-1 differentiation, the transcriptional change ($49 \pm 4\%$) was the single best predictor, followed by protein synthesis rate ($46 \pm 7\%$) and degradation rate ($15 \pm 3\%$), for the experimentally confirmed protein expression change. Combining all three parameters gives a very respectable predictive power of $69 \pm 7\%$ for any given replicate and this could be increased to 74% by simply taking the mean of the parameters from the biological replicates (Supplementary Figure S4). A breakdown of the contributions of the individual parameters in this model using one component is as follows: synthesis rate—41%, transcriptional change—45% and degradation rate—14% (Figure 5C; Supplementary Table S6), which indicates that the synthesis rate contributes more than the degradation rate to the variance not explained by the transcription change. Similarly, we observed that during C2C12 differentiation the synthesis rate predicted the protein expression change better than the degradation rate of the proteins (Supplementary Figure S5). Taken together, this clearly shows that both the synthesis and degradation rates of a protein are important processes in regulating the protein's expression change. The effect of post-transcriptional regulatory mechanisms such as microRNA after 48 h differentiation can be monitored by comparing the prediction of protein expression changes from only transcriptional changes or from transcriptional changes plus the synthesis rates. Interestingly, we observe that post-transcriptional regulation seems to fine tune the precise control of proteins expression (Supplementary Figure S6), since many proteins displayed rather small changes, which is in excellent agreement with proteomics studies measuring the effects of miRNA (Baek *et al*, 2008; Selbach *et al*, 2008).

To investigate whether proteins with increasing or decreasing expression changes were regulated similarly, we performed partial least square regression on the proteins significantly increasing or decreasing during differentiation. This revealed that the transcription change and synthesis rate

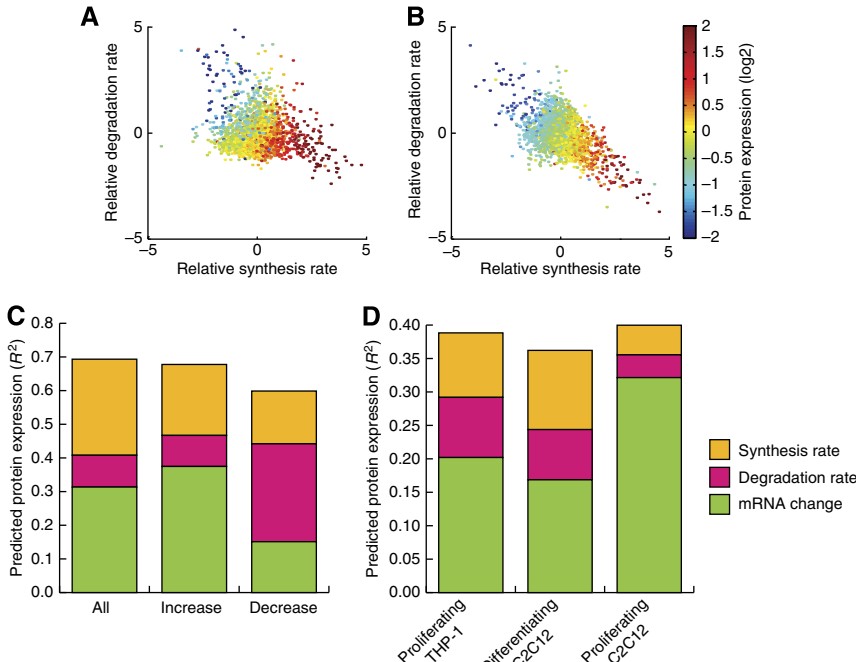

**Figure 5** Parameters controlling protein expression changes during cellular differentiation. (**A**) A scatterplot of the synthesis and degradation rates in THP-1 after 48 h of differentiation from monocytes to macrophages, with the resulting change in overall expression encoded by color. (**B**) A scatterplot of the synthesis and degradation rates in C2C12 after 48 h of differentiation from myoblasts to myotubes, with the resulting change in overall expression encoded by color. (**C**) Protein expression change predictions of all proteins, significant increasing and significant decreasing proteins, respectively derived from partial least squares (PLS) regression of the various parameters controlling protein expression change during THP-1 differentiation, using the mean of at least two out of three biological replicates per parameter. (**D**) Predicted protein expression change during THP-1 differentiation derived from PLS regression of the contributions of the various parameters controlling expression change using the synthesis and degradation rates deriving from proliferating THP-1 cells, differentiating C2C12 cells and proliferating C2C12 cells, respectively. For (**C**, **D**) a value of 1 would indicate that the model perfectly predicts the expression change.

makes the biggest contribution for proteins whose expression increases (55 and 31%, respectively), whereas the degradation rate accounted for 13% of the variance explained by the model (Figure 5C). The opposite was observed for the proteins whose expression decreased, since here the transcription change and synthesis rate accounted for only 25 and 26%, whereas the degradation rate accounted for 49% of the variance explained by the model (Figure 5C).

Next, we investigated whether the addition of degradation and synthesis rates obtained under a different cellular state could improve the prediction of protein expression changes from transcriptional changes. We therefore used the degradation and synthesis rates obtained in proliferating THP-1 cells to try to predict the protein expression in differentiating THP-1 cells. Interestingly, if one can use both the synthesis and the degradation rates in such a prediction, then the calculated changes in protein expression come much closer to the measured values (Figure 5D). Since we earlier observed a high conservation between degradation rates in the mouse and human cells, we next investigated whether knowledge of the synthesis and degradation rates in differentiating or proliferating murine C2C12 cells would improve the predictive power versus RNA expression change alone. Indeed, both the synthesis and degradation rates improve the model considerably in differentiating murine C2C12 cells, although the effect was much more modest in proliferating murine C2C12 cells (Figure 5D).

## Protein synthesis rate is intensively regulated in differentiating cells

How the synthesis and degradation rates for individual proteins respond to external perturbation has long been an open question that our data now allow us to address directly. A comparison of the synthesis and degradation rates in differentiating versus proliferating cells reveals a significantly poorer correlation for synthesis rates than for relative degradation rates in both THP-1 cells and C2C12 (Figure 6A and B). To investigate this regulation more in detail, we performed 2D enrichment analysis ($P < 0.05$) for the different biological processes between the relative synthesis rates of differentiating and proliferating cells (Figure 6C; Supplementary Figure S7). This clearly revealed that many biological processes have different synthesis rates in proliferating and differentiating cells as can be seen by their off-diagonal location. For example, we observed a decrease in the relative synthesis rates of proteins involved in DNA condensation and cell cycle in differentiating versus proliferating cells (see Supplementary Table S7 for complete list). On the contrary, the degradation rates between proliferating and differentiating cells were highly correlated in both THP-1 and C2C12 cell lines, suggesting that differentiation has little impact on degradation. 2D enrichment analysis ($P < 0.05$) of degradation rates between differentiating and proliferating cells (Figure 6D; Supplementary Figure S8) revealed a very different picture from that of the synthesis rates, since most enriched biological processes were located on

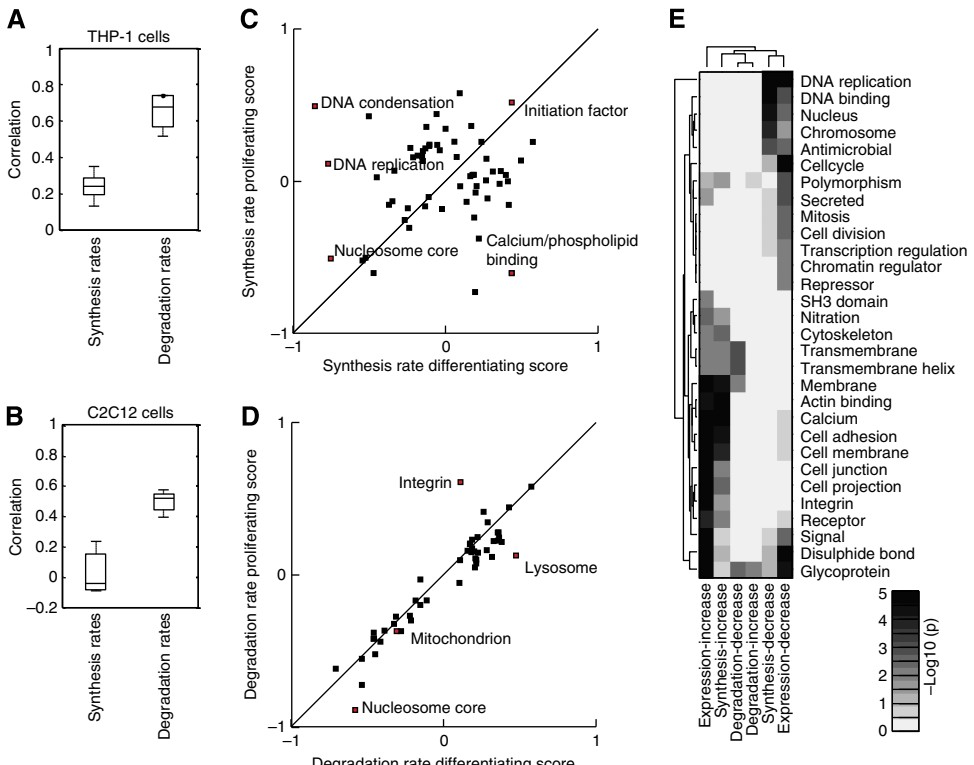

**Figure 6** Regulation of protein synthesis and degradation rates for differentiating and proliferating cells. Boxplots of the spread of correlations (Spearman) from three biological replicates between the synthesis and degradation rates in differentiating and proliferating (**A**) THP-1 and (**B**) C2C12 cells. 2D analysis ($P < 0.05$) of the functional classes of proteins enriched in different regions of proliferating versus differentiating space of THP-1 cells for (**C**) synthesis and (**D**) degradation rates. (**E**) A Clusterogram of the $-\log_{10}(P$-value) from enrichment analysis of the proteins whose expression, synthesis and degradation rates changed significantly ($P < 0.05$) during THP-1 differentiation from monocytes to macrophages. The cellular processes enriched among those proteins with increased synthesis rates also tended to be enriched in those with increased protein expression change, whereas virtually no classes were enriched among those with altered degradation rates.

the diagonal, suggesting that the degradation rates of the different biological processes are not different in the differentiating and proliferating cells (see Supplementary Table S8 for complete list). As an example, the biological process 'cell cycle' was highly affected by differentiation in THP-1 cells and if one examines the proteins assigned to this class in detail, it is obvious that they have very different synthesis rates but very similar degradation rates (Supplementary Figure S9). Lastly, a comparison of the biological processes enriched among those proteins that displayed significant changes in synthesis or degradations rates versus those that displayed significant changes in protein expression during THP-1 differentiation revealed that similar processes were enriched in those proteins with significantly changed synthesis rates and protein expression but not degradation rates (Figure 6E; Supplementary Table S9). Taken this together strongly suggests that the cell changes a protein's synthesis rate up or down either increase or decrease the amount of that protein.

## Discussion

The regulation of protein expression change in response to external stimuli is fundamental to survival of all living organisms yet, to our knowledge, no quantitative assessment of the contributions that various cellular processes make to

such a response has been made. Here, we set out to model how protein expression change is regulated during differentiation using two unrelated model systems so as to support more generalized conclusions. That unstructured, lower abundance proteins were most dramatically affected by differentiation seems to be designed to allow very fast regulation of a large part of the signal transduction network. This is also consistent with observations that the differences among cell types are largely a result of lower abundance proteins (Lundberg *et al*, 2010) and that intrinsically unstructured proteins have more interaction partners and are involved in cellular signaling (Uversky *et al*, 2005; Babu *et al*, 2012).

Measuring the synthesis and degradation rates of proteins on a system level revealed that protein assigned to the same macromolecular complex displayed similar synthesis and degradation rates. However, closer examination of the synthesis and degradation rates for components of sub-complexes revealed that the alpha-subunits of the proteasome core particle are being formed wholly from newly synthesized proteins rather than being re-assembled from recycled subunits, which implies that the proteasome core particle is also degraded as an intact complex. Thus, we speculate that the established modularity of macromolecular complexes (Gavin *et al*, 2006) is also present during synthesis and degradation of the complexes, since the different modules of a complex will display similar synthesis and degradation rates,

thereby adding an additional layer to the established assumption that proteins display different synthesis and degradation rates in different cellular compartments (Lam *et al*, 2007; Boisvert *et al*, 2012). This would also mean that synthesis and degradation rates of a protein could be used as discriminatory parameters to improve the assignment of protein interactions, similarly to mRNA co-expression (Jansen *et al*, 2002).

Toward our primary goal of modeling the contributions various cellular processes make to an eventual proteome, predictions of protein expression changes can be significantly improved by taking into account both the synthesis and the degradation rates of the proteins, contributing 41 and 13% respectively to the variance not explained by changes in transcription. That post-transcriptional regulation of synthesis and degradation rate is having such an impact after 48 h of differentiation, where the highest correlation between transcriptome and proteome expression changes was observed, suggests that the factors are even more important earlier in differentiation when RNA and protein levels are very poorly correlated. The importance of post-transcriptional regulation of protein expression changes during THP-1 differentiation was illustrated in a recent paper that describes how over-expressing four miRNAs lead to partial differentiation (Forrest *et al*, 2009). Our results found that synthesis rate contribute more than degradation rate to the prediction from transcript to protein expression changes which is in agreement to what has been observed for these processes' contributions in predicting protein abundance, where it was observed that protein synthesis rate was a better predictor than protein degradation rate (Schwanhäusser *et al*, 2011).

We observed clear differences in the extent to which synthesis and degradation rates contributed to the changes of protein expression during differentiation; perhaps unsurprisingly the synthesis rate was the best predictor for those proteins that increased in expression whereas the degradation rate contributed the most in predicting decreasing proteins. This does suggest though that failure to consider degradation rates is the most likely explanation for why a poor correlation is generally observed between transcriptome and proteome expression changes for proteins whose expression decreases (Lee *et al*, 2011). That we observe a strong correlation between degradation rates in proliferating and differentiating cells suggests that one does not even have to measure the degradation rate in the same cellular state of the system of interest in order to take such a parameter into account when using transcriptome changes to predict protein expression changes; degradation rates are now being recorded in the newly developed Encyclopedia of Proteome Dynamics (Larance *et al*, 2013).

Perhaps, the most striking finding here is that the changes in any given protein's expression during differentiation are largely due to changes in the synthesis rate, while the degradation rates remain constant. Using the bathtub analogy from the introduction, this means that the water level in the bathtub will be controlled mainly from the faucet either by increasing or by decreasing the flow, whereas the draining will be at a nearly constant flow (Figure 7A and B). This has at least two conceptual consequences: (1) the drain size for an individual protein is likely to vary little, that is, the drain size is less adjustable than the faucet and; (2) the faucet has a key role in preventing overflows. At the protein level, consequence 1 translates into a strong 'default' component in the degradation of many proteins, which we propose is based on the protein's sequence and can be either dependent or independent of the ubiquitylation (Tasaki *et al*, 2005, 2013; Asher *et al*, 2006). Consistent with this are previous observations that many proteins involved in signal transduction and transcriptional regulation have fast basal turnover rates and, consequently, they can be cleared quickly in response to perturbation; this would be energetically favorable to the cell since these proteins are typically expressed at low levels and therefore less energy would need to be consumed (Gsponer *et al*, 2008; Legewie *et al*, 2008). Interestingly, the regulation of changes in mRNA levels is also predominantly regulated at the level of transcription while mRNA degradation rate is generally constant (Rabani *et al*, 2011). Even though we state here that a change in synthesis rate is the main driver of expression change during differentiation, this does not exclude an important role for the ubiquitin-proteasomal system. The degradation rate in differentiating and proliferating cells does not correlate perfectly, meaning that the degradation rates of some proteins are regulated. Furthermore, differentiation is a relative slow process so perhaps changes in degradation rates could have a larger role in regulating protein expression changes in response to acute stimuli such as heat shock or inflammation (Bhoj and Chen, 2009; Fang *et al*, 2011). Finally, since this study only focused on long-term changes of the synthesis and degradation rates it could very well be that changes in degradation rates have a more important role earlier in differentiation.

Consequence 2 means that a tight regulation of the synthesis of proteins is a key to prevent dangerously high and potentially toxic levels of some proteins (Figure 7C). In agreement with this are previous observations showing that accumulation of malfolded proteins causes dephosphorylation of eIF2α, which leads to repression of translation (Jiang and Wek, 2005; Moreno *et al*, 2012). Furthermore, it has been observed that mTOR gets sequestered in polyglutamine aggregates during

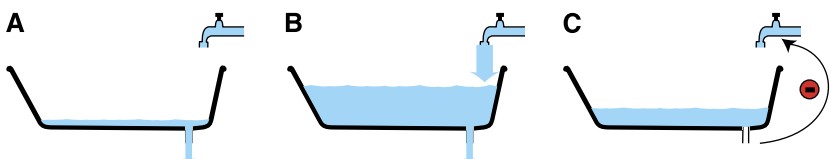

**Figure 7** Model of protein expression change. (**A**) Proteins decreasing in expression are a result of decreased protein synthesis rate, whereas the protein degradation rate is constant. (**B**) Proteins increasing in expression are a result of increased protein synthesis rate, whereas protein degradation rate is constant. (**C**) Inhibition of the protein degradation rate will lead to feed back inhibition of the synthesis rate.

Huntington's disease, resulting in decreased mTOR-dependent TOP translation and introduction of autophagy, thereby serving a protective function (Ravikumar *et al*, 2004).

These findings then raise the captivating question: why has evolution favored regulation of synthesis rather than of degradation? One possibility is that altering degradation rates might too easily induce protein aggregation, similar to what is seen when the autophagy or the proteasome is inhibited (Hara *et al*, 2006; Komatsu *et al*, 2006; Bence *et al*, 2001). One way the cell avoids aggregate formation is by keeping the lifetimes of aggregate-prone proteins short (De Baets *et al*, 2011; Gsponer and Babu, 2012), which would be more difficult if the lifetime of a protein could be extended in response to perturbation.

# Conclusion

Here, we have demonstrated that changes in protein synthesis rates are the primary drivers of differentiation. Clear and consistent trends in support of this were observed in very diverse cell types from two different organisms, providing strong evidence that these trends are universal characteristics of cellular differentiation. Furthermore, we have provided quantitative evidence to support the common assumption that the reason why transcriptomes and proteomes frequently correlate very poorly is that there is still substantial variance imparted at the levels of protein synthesis and degradation. This can even be observed in sub-populations of a given protein, such as those bound in specific protein complexes. Future experiments will hopefully extend these measurements to other systems to test how widely these characteristics are conserved.

# Materials and methods

## Identifying changes in the proteome during differentiation of THP-1 and C2C12 cells

THP-1 or C2C12 cells were grown in RPMI or DMEM media, respectively, with added 10% dialyzed fetal bovine serum (FBS), 1% glutamine, 1% non-essential amino acids, 1% penicillin/streptomycin and (L-[U-$^{13}C_6$, $^{14}N_4$]arginine and L-[$^2H_4$]lysine or L-[U-$^{12}C_6$, $^{14}N_4$]arginine [$^1H_4$]lysine or L-[U-$^{13}C_6$, $^{15}N_4$]arginine and L-[U-$^{13}C_6$, $^{15}N_2$]lysine (Cambridge Isotope Labs, Cambridge, MA). The cells were grown for at least five doublings to ensure 100% incorporation of labeled amino acids before THP-1 cells were differentiated by 25 nM PMA and C2C12 cells were differentiated by increasing the confluency to 100% while decreasing the serum concentration to 2%.

## Peptide separation and MS

The cells were washed three times in PBS and lysed in 1% deoxycholate before being boiled for 5 min. The lysate was digested to peptides as in Rogers and Foster (2007) before being separated by IEF (Agilent Technology) following the manufacturers' instructions. The separated peptides were STAGE Tipped as in Rappsilber *et al* (2007) before being analyzed by MS as in Kristensen *et al* (2012) and proteins were identified and quantified using MaxQuant (Cox and Mann, 2008) with the settings as supplied in Supplementary methods. The mass spectrometry data acquired here have been deposited to the ProteomeXchange Consortium (http://proteomecentral.proteomexchange.org) via the PRIDE partner repository (Vizcaíno *et al*, 2013) with the data set identifier PXD000328.

# Data analysis

Significantly changing proteins were identified by applying ANOVA between the five time points with the following settings (permutation-based FDR, $P = 0.05$, S0 = 1, 250 randomizations) using Perseus. Increasing and decreasing proteins were defined by clustering the data into two clusters using fuzzy C mean clustering. Wilcoxon–Mann–Whitney test, partial least regression, and KEN-box motif determination were performed using Matlab (http://www.mathworks.com), whereas correlation coefficients and 2D enrichment analysis ($P < 0.05$, Benjamini-Hochberg FDR for truncation) of the biological processed (Uniprot Keywords) were performed in Perseus, similarly to Cox and Mann (2012). In boxplots, points were drawn as outliers if they were larger than $q_3 + 1.5$ $(q_3 - q_1)$ or smaller than $q_3 - 1.5$ $(q_3 - q_1)$.

Finally, enrichment (Uniprot Keywords) analysis of changing proteins, synthesis and degradation rates were performed by GPROX (Rigbolt *et al*, 2011) using Fisher's exact test ($P < 0.05$, Benjamini-Hochberg FDR for truncation).

For further detail, see Supplementary methods.

# Supplementary information

# Acknowledgements

We wish to thank Nikolay Stoynov and Joost Gouw for assistance with the mass spectrometers used in this study, Nat. F. Brown for producing Figures 1A and 7, and the PRIDE Team for help with depositing the data. An Operating Grant from the Canadian Institutes of Health Research (MOP-77688) to LJF supported this research. ARK was supported by a four-year fellowship from the University of British Columbia and by the Danish Agency for Science Technology and Innovation. LJF is the Canada Research Chair in Quantitative Proteomics. The mass spectrometry infrastructure used here was supported, in part, by funds from the Canada Foundation for Innovation, the BC Knowledge Development Fund and the BC Proteomics Network.

*Author contributions:* ARK conceived of and performed the experiments; ARK, JG and LJF analyzed the data and wrote the manuscript.

# Conflict of interest

The authors declare that they have no conflict of interest.

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
