## [Review Process File · Molecular Systems Biology]

Protein synthesis rate is the predominant regulator of protein expression during differentiation

Anders R Kristensen, Joerg Gsponer and Leonard J Foster

Corresponding author: Leonard Foster, University of British Columbia

Review timeline:

Submission date:	24 April 2013
Editorial Decision:	21 June 2013
Revision received:	24 July 2013
Editorial Decision:	19 August 2013
Revision received:	21 August 2013
Accepted:	21 August 2013

Editors: Maria Polychronidou

Transaction Report:

1st Editorial Decision

21 June 2013

Thank you again for submitting your work to Molecular Systems Biology. We have now heard back from the three referees who agreed to evaluate your manuscript. As you will see from the reports below, the reviewers acknowledge that your work is addressing a potentially interesting topic. However, they raise a series of concerns, which should be carefully addressed in a revision of the manuscript.

Without repeating all the points listed below, the more fundamental issues refer to the need to convincingly demonstrate the validity and the generality of the major conclusions. Please refer to the comments of reviewers #1 and #3 for a detailed description of the points that need to be addressed.

On a more editorial level, as the reviewers have suggested, the manuscript needs to be carefully rewritten in order to make the key findings and major conclusions easily accessible. Moreover, I would like to ask you to deposit the proteomics data in the appropriate public databases. (Additional information is available in the "Guide for Authors" section in our website at <http://www.nature.com/msb/authors/index.html#a3.5.2>.) Please include the links and accession numbers in the "Data Availability" section of your manuscript.

If you feel you can satisfactorily deal with these points and those listed by the referees, you may wish to submit a revised version of your manuscript. Please attach a covering letter giving details of the way in which you have handled each of the points raised by the referees. A revised manuscript will be once again subject to review and you probably understand that we can give you no guarantee at this stage that the eventual outcome will be favorable.

REFeree REPORTS

Reviewer #1

The authors employ a metabolic labeling-based approach to explore protein synthesis and degradation rates in two cell-based differentiation models: the THP1 monocyte cell system and the C2C12 myoblast cell system. In the first part of the manuscript, they characterize their systems, largely confirming previous observations, for instance regarding the effect of protein disorder and protein abundance on protein stability. They next perform PCP SILAC and size exclusion chromatography to explore the relative synthesis/degradation rates for protein complex components. They next investigate the correlation between transcriptome and proteome, and model the control of protein expression to finally conclude that protein synthesis is intensively regulated in differentiating cells.

Perhaps the most important contribution here is the generation of the dataset itself: there is currently very little data available that provide global information regarding the rates of synthesis and degradation in mammalian cell systems, especially during processes such as differentiation, which is what is being done here. As such, the entire mass spectrometry data should be made fully available, alongside proper experimental annotation, perhaps via the ProteomeXchange consortium.

The analytical parts of the manuscript are in general less interesting, perhaps due to the large number of different (and often confirmatory) points that the authors are making throughout. The presentation of the results is confusing, which makes it very difficult to truly evaluate the quality/novelty/significance of the manuscript. Often times, the authors are making fairly vague statements, which should be better quantified and described. In particular, the novelty aspects should be better stated here.

If the manuscript were to be considered for publication, I would recommend a major effort at rewriting the it (and making it much shorter), and increasing the quality/clarity of the figures and tables.

There are also other issues with the manuscript:

1) The analysis of co-regulation of turnover rates for proteins in complexes seems like a different (disconnected) story as presented. Firstly, I am not sure that this should be part of the same manuscript, as there are already many lines of investigation in it. Secondly, it is unclear what are the major conclusions here. A few examples are presented (essentially, proteasome components and the PKA complex), but it is not clear whether these are isolated cases or whether "the synthesis and degradation rates on the sub-complex level provides important modular information on the composition of the macromolecular complex". For how many different complexes / complex components have these kinds of analyses been possible? How often are there differences at the level of synthesis? Degradation? Intuitively, one would suspect that proteins which are not associated with their natural binding partners may be more subjected to degradation. Does this hold true here? Are there ways to corroborate these observations by an orthogonal approach? It seems like this part of the manuscript is too preliminary at this point.

2) As the title indicates, the key point of the manuscript may be that the protein synthesis (rather than degradation) rates are key regulators of protein expression (besides transcriptional changes) during differentiation. However, this conclusion gets drowned here through all the parallel stories that the authors are developing. Again, this would be nice if the authors could validate some of their data, but also develop their points more in depth. Are we looking at changes in translation due to microRNA regulation? Structures in the 3' and 5'UTR?

3) Sup Figure 1 is a bit confusing. First, the labels don't really align (perhaps color code the desired nodes and labels?). Second, it may be useful to mark the areas of confident change, or at least label the axes differently (i.e. centered on the 0,0 point). How do these categories (and individual proteins) compare to the changes in mRNA levels detected upon differentiation? Note that the somewhat cosmetic changes suggested here may also improve the understanding of Figures 2E and

4C.

4) It was very difficult to make sense of the supplementary tables given the information provided. For example, the authors refer on page 5 to "hundreds of potentially new markers", and then go on to define which of the identified proteins are changing. However, this is not very clear what they mean, and which of the identified (and presumably quantified) proteins are good potential markers. Perhaps the authors should consider adding information regarding their cutoff for changes directly on those Excel files? As it is, these are not tremendously useful. It would also be nice if the GO categories, etc., could somehow be merged onto the source Excel files. Lastly, providing a short description of the content of each Excel file - in addition to the column descriptors, would have been helpful.

5) The comparisons to the mRNA level data can only be carried so far, given that the experiments are not performed here, but simply extracted from the literature. While I realize that this is a common practice, there are certainly limitations to the approach, especially for processes such as the differentiation system used by the authors. What is the variability in the differentiation process and how could this influence the results presented here?

6) I am not completely convinced that much can be done regarding the analysis of human versus mouse cell lines in terms of conservation of degradation through amino acid similarity, as the authors suggest on page 8. The authors are using a single mouse (C2C12) and a single human cell line (THP1), which are from different cellular origin. Of course, their conclusions are largely in agreement with previously published data, but I think you could make their correlation (0.65) fit any conclusion. In fact, if they simply moved to the discussion of Figure 2E and rework it to say that there are commonalities in the protein processes regulated by degradation across two fairly different cell types (and, yes, species), their conclusions could be much stronger.

7) I could not find "Table 1", mentioned on page 5.

8) The changes in abundance for CD11B and C appear very modest. Was this expected?

Reviewer #2

Kristensen et al have studied the contribution of protein synthesis and degradation rates to overall protein expression levels using SILAC labelling and quantitative mass spectrometry. Previous studies have used similar methods to make conclusions on this question, however, this manuscript differs from other papers in 2 key aspects:

- (i) The authors have made these measurements over a time course perturbation (differentiation) which facilitates conclusions on the mechanisms used by cells to alter overall protein expression levels in response to a changing environment as opposed to at steady state.
- (ii) By incorporating native separations in the form of size exclusion chromatography the authors were able to incorporate information regarding the composition of protein complexes, and thereby determine whether protein synthesis/degradation rates were related to protein complex (or complex subunit) membership.

The primary conclusions that arise from the study are firstly, that the primary driver of overall protein levels during cellular differentiation is protein synthesis, and secondly that members of a protein complex/subcomplex share similar synthesis/degradation rates (with certain exceptions that may point toward interesting biology e.g. participation of certain proteins in multiple complexes, etc). These are interesting and relevant findings and should prove to be of general interest to the systems biology community. The manuscript seems technically sound and is clearly written. There are some clarifications required. If the points below can be satisfactorily addressed then publication would be recommended.

1. At many points in the manuscript the authors state that the protein synthesis rate is the primary driver in overall protein abundance changes during differentiation and this statement seems to be well supported by the data they present. Speculation that this mechanism applies generally to other perturbations (especially perturbations on a similar time scale to differentiation) is tempting and probably reasonable but is not directly supported by the data in the manuscript. Regulation of key protein levels in perturbed systems by strong degradation is well described in some systems, such as the classical example of p53 in DNA damage. As such, generalizing to any perturbation, and

especially for perturbations at faster time scales is risky. The authors touch on this in the discussion ("... acute stimuli ...") but it seems a more balanced discussion of the potential to generalise these results is warranted.

2. Extending from point 1, did the authors find any proteins that were exceptions to the general trend whose levels were strongly degradation driven during the perturbation (in the early time point perhaps) which might point to specialised mechanisms of regulation?

Reviewer #3

Kristensen et al. used quantitative proteomics to monitor protein expression changes during cell differentiation in a human cell line. In principle the study is important as the contributions of protein synthesis and degradation to protein level changes are poorly studied. However, before acceptance of this manuscript can be considered, the major concern below must be successfully addressed. The minor comments may serve to improve the manuscript.

Major concern:

In the abstract and main text the authors say: "the cell changes a protein's synthesis rate up or down either to increase or decrease the amount of that protein" and later in the discussion: "Perhaps the most striking finding here is that the changes in any given protein's expression during differentiation are largely due to changes in the synthesis rate, while the degradation rates remain constant". This statement appears inconsistent with Fig. 5 where, unless I totally misunderstood, it is clearly visible that the degradation rates are not constant but vary in murine cells in the range observed for the synthesis rates. Fig. 5C shows that for decreasing protein levels in differentiating cells the degradation rate has a higher predictive power suggesting that degradation is the predominant mechanism for proteins with decreasing expression. If I understand correctly, the only basis for the conclusion that synthesis rate is the predominant regulator is the correlation plot in Fig. 6A,B. The authors should clarify why they believe that the relative changes seen in Fig. 5 are not important and do not contribute to a protein expression level change.

Minor comments:

1. Quantitative proteomics reveals mechanisms behind phenotypic changes:

"In addition, those proteins altered by differentiation tended to be more intrinsically disordered ($P=0.03$, Wilcoxon-Mann-Whitney test) (Fig. 1e) and were generally lower in abundance ($P=2.8 \cdot 10^{-5}$, Wilcoxon-Mann-Whitney test) than the unregulated proteins (Fig. 1f)."

I recommend to leave out Figure 1e and 1f. The observations are not very significant, and the results presented in 1f appear not very solid due to the statistical testing approach used. Also, the figure panels do not give much additional information. The disorder of proteins is shown to play a significant role in stability in Figure 2c anyway, leading to redundancy.

2. Turnover rates of macromolecular sub-complexes:

"Proteins within a complex are vastly ($P<10^{-99}$, Wilcoxon-Mann-Whitney test) closer in Euclidian space than the rest of the proteins (Fig. 3a)."

Is this only apparent for the combination of synthesis and decay rates or could it also be explained by one of the opposing mechanisms. If not, which of the rates has a stronger influence? This must be addressed and at least discussed as the authors otherwise state that synthesis plays the predominant role.

3. Temporal correlation between the transcriptome and the proteome:

This referee has concerns about the figures: please label all figures in order to clarify the results. E.g. add legends, add information to what is correlated with what for the correlation plots. Generally, for correlation and box plots, please indicate the number of data points for error assessments.

4. Figure 4A: labeling of axis and description is not clear. Of how many data points were the box plots made of? Please better show scatterplots of averaged replicates.

5. "When we examined the correlation of mRNA and proteins expression change at each of the five

time points, we observed that the relationship nearly reached a steady state after 24h differentiation, suggesting that extensive post transcriptional regulation is taking place during early differentiation (Fig. 4b)."

If I understood this correctly the mRNA and protein levels compared are each measured at the same time points. In a dynamic adaptation to a certain stimulus, you would however in any case expect a lag between changed mRNA levels and their translation if the steady state is not reached. How can the authors conclude extensive post-transcriptional regulation? This should be deleted or clarified with solid arguments.

6. "with less than 6 % of the genes showing opposite regulation at 48 hours differentiation" - this must be clarified.

7. Modeling the control of protein expression:

"If one looks first at the effect of synthesis and degradation rates on overall protein expression, it is obvious that if the synthesis and degradation rates of a protein are equal then the expression of the protein is stable (Fig. 5a-b)."

If I understood this correctly the word stable is used incorrectly in this context. I think the authors mean "steady state levels". Please clarify.

8. Fig. 5a/b: What are the conclusions from these panels? Are these data anti-correlated? If yes this is a nice finding because it shows that cells synthesize proteins they need for differentiation and at the same time stabilize them. Proteins that are not needed are not synthesized any more and also show higher degradation rates. If the authors agree I recommend emphasizing this.

9. Fig. 5d is not clear to me. I do not see an increase in the predictive power of synthesis or degradation rates and I cannot see which part of the text belongs to which part in the figure. This must be clarified or taken out.

1st Revision - authors' response

24 July 2013

Responses to reviewers:

Reviewer #1

The authors employ a metabolic labeling-based approach to explore protein synthesis and degradation rates in two cell-based differentiation models: the THP1 monocyte cell system and the C2C12 myoblast cell system. In the first part of the manuscript, they characterize their systems, largely confirming previous observations, for instance regarding the effect of protein disorder and protein abundance on protein stability. They next perform PCP SILAC and size exclusion chromatography to explore the relative synthesis/degradation rates for protein complex components. They next investigate the correlation between transcriptome and proteome, and model the control of protein expression to finally conclude that protein synthesis is intensively regulated in differentiating cells.

Perhaps the most important contribution here is the generation of the dataset itself: there is currently very little data available that provide global information regarding the rates of synthesis and degradation in mammalian cell systems, especially during processes such as differentiation, which is what is being done here. As such, the entire mass spectrometry data should be made fully available, alongside proper experimental annotation, perhaps via the ProteomeXchange consortium.

RESPONSE: We thank the reviewer for his/her kind comments and are very satisfied that s/he finds it is an important contribution. We certainly do not object to sharing the underlying data and have now made it available via ProteomeXchange. This can be accessed using Pride inspector (<http://www.ebi.ac.uk/pride/>), Username: review53570; Password: JThrebTw.

We have as well included the following to the Materials and Method section:
p. 18 The mass spectrometry data acquired here have been deposited to the ProteomeXchange Consortium (<http://proteomecentral.proteomexchange.org>) via the

PRIDE partner repository (Vizcaíno et al, 2013) with the dataset identifier PXD000328.

The analytical parts of the manuscript are in general less interesting, perhaps due to the large number of different (and often confirmatory) points that the authors are making throughout. The presentation of the results is confusing, which makes it very difficult to truly evaluate the quality/novelty/significance of the manuscript. Often times, the authors are making fairly vague statements, which should be better quantified and described. In particular, the novelty aspects should be better stated here.

We agree with this reviewer that the novelty could have been better stated; therefore we have included the following statements:

p. 8 The discovery approach to investigating the synthesis and degradation used here allowed us, for the first time, to unravel if any relationship exists between protein complex size and the degradation and synthesis rates of the components.

p. 9 ... we are able, for the first time, to capture that the proteasome alpha-ring gets de-novo assembled from only newly synthesized proteins.

p. 9-10 ... suggesting for the first time that both the ubiquitin receptors RPN10 and RPN13 can exchange with their free forms and that this approach provides a completely novel ability to probe such details.

p. 11 This suggests a novel concept: that if protein expression changes can be measured, then they could be used to make much more accurate predictions of gene expression changes than what mRNA expression changes alone would yield.

p. 11 We then applied partial least square regression to our own data from differentiating THP-1 cells and similar transcriptional changes data from Suzuki et al (Suzuki et al, 2009) to examine for the first time the contributions of various factors across the board of all proteins to the ultimate changes in protein expression.

If the manuscript were to be considered for publication, I would recommend a major effort at rewriting the it (and making it much shorter), and increasing the quality/clarity of the figures and tables.

We agree with this reviewer that there are many confirmatory points; indeed, that was intentional since we wanted to make the point that our approach was providing very reliable data consistent with other reports. Since none of the reviewers expressed any doubts with the data quality, we have undertaken a major effort in rewriting the manuscript by cutting some of the confirmatory points, and making the figures more clear, which have made the manuscript shorter and as well stressed the novelty aspects. This led us to drop the original 2a-d, 3a and 3e.

There are also other issues with the manuscript:

1) The analysis of co-regulation of turnover rates for proteins in complexes seems like a different (disconnected) story as presented. Firstly, I am not sure that this should be part of the same manuscript, as there are already many lines of investigation in it. Secondly, it is unclear what are the major conclusions here. A few examples are presented (essentially, proteasome components and the PKA complex), but it is not clear whether these are isolated cases or whether "the synthesis and degradation rates on the sub-complex level provides important modular information on the composition of the macromolecular complex". For how many different complexes / complex components have these kinds of analyses been possible? How often are there differences at the level of synthesis? Degradation? Intuitively, one would suspect that proteins which are not associated with their natural binding partners may be more subjected to degradation. Does this hold true here? Are there ways to corroborate these observations by an orthogonal approach? It seems like this part of the manuscript is too preliminary at this point.

We agree with the reviewer that the way the section about protein complexes was presented

could seem like it was a disconnected story. The section served as 1) a validation that RPN10 was having different synthesis/degradation rates than the regulatory particles proteins, 2) To investigate if the core subunit of the proteasome as well was showing similar synthesis/degradation rates on the sub-complex level.

We decided to perform a discovery-based experiment (versus targeted), since we would be able to measure the synthesis and degradation rates of the proteins we were interested in (the proteasome) but for the same cost we could also generate a larger dataset that could provide important data for other researchers. We have therefore decided to not do a detailed analysis of this data, since that would make this aspect really a completely disconnected story from the rest of the manuscript.

Based on this reviewer's recommendation though we have decided to make it clearer that the data presented here mainly served as a validation and detailed analysis of the proteasome sub-complexes. Therefore we have deleted the section about PKA, figure 3A and 3E and shortened the text to exclude some confirmatory points.

2) As the title indicates, the key point of the manuscript may be that the protein synthesis (rather than degradation) rates are key regulators of protein expression (besides transcriptional changes) during differentiation. However, this conclusion gets drowned here through all the parallel stories that the authors are developing. Again, this would be nice if the authors could validate some of their data, but also develop their points more in depth. Are we looking at changes in translation due to microRNA regulation? Structures in the 3' and 5'UTR?

More than one reviewer made a similar point, that there were too many apparently disconnected stories, so we cut figure 2A-D out and eliminated some of the confirmatory points. We are a little surprised about this reviewer's point about the lack of validation, however, since a key point of including a second cell line was for its validation value. To our minds this seems like a better approach to validation than trying to pick one protein or complex and validate it using antibodies, shRNA or other similar tools. By using a second cell line we are able to show that the trends we are seeing are consistent for many different aspects and different proteins or complexes and we hope that the reviewer will agree with us in this regard

To the role of microRNA and structure in the 3' and 5' UTR, we have included supplementary figure 6 showing the effect of post-transcriptional regulation. Our data points towards this effect serving primarily as fine tuning mechanism, since there is extremely good correlation between mRNA and protein expression after 48 hours differentiation (only seven proteins and genes shows significant opposite regulation). Hereby it is very difficult to do any analysis based on these seven proteins. Furthermore we analyzed if the protein-mRNA expression was further away from the diagonal for genes which miRNA have been shown to be important for THP-1 differentiations (Forrest et al, 2009). This was not the case for any of the miRNA tested (see below), which is probably to be expected since there is presumably a combinatorial effect of these and other miRNAs. However since we cannot predict what those combinatorial effects might be we have not included this analysis in the manuscript.

	all	mir155	mir222	mir424	mir503
Median distance from diagonal	0.3021	0.3012	0.3351	0.1543	0.2675
stdev	0.4073	0.3662	0.4027	0.3221	0.2961
Number of targets in our dataset		72	60	9	46
p-value (Wilcoxon-Mann-Whitney test)		0.987	0.5083	0.2598	0.6829

3) Sup Figure 1 is a bit confusing. First, the labels don't really align (perhaps color code the desired nodes and labels?). Second, it may be useful to mark the areas of confident change, or at least label the axes differently (i.e. centered on the 0,0 point). How do these categories (and individual proteins) compare to the changes in mRNA levels detected upon differentiation? Note that the somewhat cosmetic changes suggested here may also improve the understanding of Figures 2E and 4C.

Based on this reviewer's and reviewer 3's comments we have reworked the 2D enrichment analysis figures so the axes align and are centered around 0,0.

4) It was very difficult to make sense of the supplementary tables given the information provided. For example, the authors refer on page 5 to "hundreds of potentially new markers", and then go on to define which of the identified proteins are changing. However, this is not very clear what they mean, and which of the identified (and presumably quantified) proteins are good potential markers. Perhaps the authors should consider adding information regarding their cutoff for changes directly on those Excel files? As it is, these are not tremendously useful. It would also be nice if the GO categories, etc., could somehow be merged onto the source Excel files. Lastly, providing a short description of the content of each Excel file - in addition to the column descriptors, would have been helpful.

We agree with the reviewer that we should have made it more clear which proteins we would consider potentially good markers between the monocytes and macrophages. We have therefore included the following statement:

p. 5 ... and found 457 proteins with significant changes in expression during differentiation, which could serve as potentially new markers for the differentiation process (see supplementary table 1 for complete list).

In addition, we have included the GO categories and Uniprot Keywords for functional enrichment analysis in supplementary table 1 as well as a short description of the content of each supplementary table.

5) The comparisons to the mRNA-level data can only be carried so far, given that the experiments are not performed here, but simply extracted from the literature. While I realize that this is a common practice, there are certainly limitations to the approach, especially for processes such as the differentiation system used by the authors. What is the variability in the differentiation process and how could this influence the results presented here?

We agree with the reviewer that it is very important when comparing mRNA and protein expression from data recorded in two different laboratories that the experimental settings are as close as possible. But as mentioned in the manuscript the cells and stimuli times are as similar as we could make them. The reported correlation is also as high as comparisons of mRNA and protein expression that have measured in the same laboratory (Lee et al, 2011) and are generally quite a bit higher than the levels of correlation reported in most studies so it would seem that inter-lab differences here have been minimized.

6) I am not completely convinced that much can be done regarding the analysis of human versus mouse cell lines in terms of conservation of degradation through amino acid similarity, as the authors suggest on page 8. The authors are using a single mouse (C2C12) and a single human cell line (THP1), which are from different cellular origin. Of course, their conclusions are largely in agreement with previously published data, but I think you could make their correlation (0.65) fit any conclusion. In fact, if they simply moved to the discussion of Figure 2E and rework it to say that there are commonalities in the protein processes regulated by degradation across two fairly different cell types (and, yes, species), their conclusions could be much stronger.

We agree with the reviewer, and have deleted figure 2A-D, since none of the reviewers expressed any concern with the method. Hereby we have simply moved forward to Figure 2E as suggested by this reviewer.

7) I could not find "Table 1", mentioned on page 5.

We apologies for this and will remember to upload it this time

8) The changes in abundance for CD11B and C appear very modest. Was this expected?

The most typical way of quantifying the expression of CD11B and C between monocytes and macrophages is by flow cytometry, where the % of positive cells is calculated. From our own estimation of the data from Prieto et al. (Prieto et al, 1994) they observe a 6 ± 1 -

and 4 ± 1 -times increase in positive cells of CD11B and C, respectively. We observe a 4.4 ± 1.2 and 5.0 ± 1.3 times (not Log2 transformed) increase in protein expression.

Reviewer #2

Kristensen et al have studied the contribution of protein synthesis and degradation rates to overall protein expression levels using SILAC labelling and quantitative mass spectrometry. Previous studies have used similar methods to make conclusions on this question, however, this manuscript differs from other papers in 2 key aspects:

- (i) The authors have made these measurements over a time course perturbation (differentiation) which facilitates conclusions on the mechanisms used by cells to alter overall protein expression levels in response to a changing environment as opposed to at steady state.
- (ii) By incorporating native separations in the form of size exclusion chromatography the authors were able to incorporate information regarding the composition of protein complexes, and thereby determine whether protein synthesis/degradation rates were related to protein complex (or complex subunit) membership.

The primary conclusions that arise from the study are firstly, that the primary driver of overall protein levels during cellular differentiation is protein synthesis, and secondly that members of a protein complex/subcomplex share similar synthesis/degradation rates (with certain exceptions that may point toward interesting biology e.g. participation of certain proteins in multiple complexes, etc). These are interesting and relevant findings and should prove to be of general interest to the systems biology community. The manuscript seems technically sound and is clearly written. There are some clarifications required. If the points below can be satisfactorily addressed then publication would be recommended.

1. At many points in the manuscript the authors state that the protein synthesis rate is the primary driver in overall protein abundance changes during differentiation and this statement seems to be well supported by the data they present. Speculation that this mechanism applies generally to other perturbations (especially perturbations on a similar time scale to differentiation) is tempting and probably reasonable but is not directly supported by the data in the manuscript. Regulation of key protein levels in perturbed systems by strong degradation is well described in some systems, such as the classical example of p53 in DNA damage. As such, generalizing to any perturbation, and especially for perturbations at faster time scales is risky. The authors touch on this in the discussion ("... acute stimuli ...") but it seems a more balanced discussion of the potential to generalise these results is warranted.

We agree with the reviewer that the discussion could be more balanced and have tried to make it so.

2. Extending from point 1, did the authors find any proteins that were exceptions to the general trend whose levels were strongly degradation driven during the perturbation (in the early time point perhaps) which might point to specialised mechanisms of regulation?

We agree with this reviewer that during early differentiation changes in the protein degradation rate could very well serve a more important role in the control of protein expression. However, in this study we have only focused on long-term changes of the synthesis and degradation rates. Together with point 1 by this reviewer we have included the following in the Discussion:

p. 17 Finally since this study only focused on long term changes of the synthesis and degradation rates it could very well be that changes in degradation rates plays a more important role earlier in differentiation.

And the following in the abstract

p. 2 As a general rule it seems that protein expression during cellular differentiation is largely controlled by changes in the relative synthesis rate, whereas the relative degradation rate of the majority of proteins stays constant.

Reviewer #3

Kristensen et al. used quantitative proteomics to monitor protein expression changes during cell differentiation in a human cell line. In principle the study is important as the contributions of protein synthesis and degradation to protein level changes are poorly studied. However, before acceptance of this manuscript can be considered, the major concern below must be successfully addressed. The minor comments may serve to improve the manuscript.

Major concern:

In the abstract and main text the authors say: "the cell changes a protein's synthesis rate up or down either to increase or decrease the amount of that protein" and later in the discussion: "Perhaps the most striking finding here is that the changes in any given protein's expression during differentiation are largely due to changes in the synthesis rate, while the degradation rates remain constant". This statement appears inconsistent with Fig. 5 where, unless I totally misunderstood, it is clearly visible that the degradation rates are not constant but vary in murine cells in the range observed for the synthesis rates. Fig. 5C shows that for decreasing protein levels in differentiating cells the degradation rate has a higher predictive power suggesting that degradation is the predominant mechanism for proteins with decreasing expression. If I understand correctly, the only basis for the conclusion that synthesis rate is the predominant regulator is the correlation plot in Fig. 6A,B. The authors should clarify why they believe that the relative changes seen in Fig. 5 are not important and do not contribute to a protein expression level change.

We believe that this reviewer may have misunderstood figure 5. Nowhere in Figure 5 or the text about figure 5 are we talking about synthesis and degradation rate changes; rather, it is about the contributions that relative synthesis and degradation rates (and mRNA changes) make to the overall protein expression changes. Figure 5A and B show the protein expression as a function of the synthesis and degradation rate, not the synthesis and degradation rate changes, and similarly the partial least square regression analysis is performed using synthesis and degradation rates not changes. During our initial preparation of the manuscript we came up with the bathtub analogy to help communicate this concept and we hope that if the reviewer re-reads this section with this in mind that it will be clearer. However, just to rule out any confusion we have included the following sentence:

p. 12 Taken together this clearly shows that both the synthesis and degradation rates of a protein are important processes in regulating the protein's expression change.

Minor comments:

1. Quantitative proteomics reveals mechanisms behind phenotypic changes:

"In addition, those proteins altered by differentiation tended to be more intrinsically disordered ($P=0.03$, Wilcoxon-Mann-Whitney test) (Fig. 1e) and were generally lower in abundance ($P=2.8 \cdot 10^{-5}$, Wilcoxon-Mann-Whitney test) than the unregulated proteins (Fig. 1f)."

I recommend to leave out Figure 1e and 1f. The observations are not very significant, and the results presented in 1f appear not very solid due to the statistical testing approach used. Also, the figure panels do not give much additional information. The disorder of proteins is shown to play a significant role in stability in Figure 2c anyway, leading to redundancy.

We do not agree with the reviewer that P-values of 0.03 and $2.8 \cdot 10^{-5}$ are not significant. They may not be as significant as they could be but the biological community typically works at a P-value cut-off of 0.05. Furthermore we believe Fig 1E-F clearly shows that the changing proteins are less abundant and more unstructured. While there isn't actually any redundancy between Figures 1F (which deals with protein expression changes) and 2C (which deals with turnover rates), in an effort to shorten the manuscript we have removed 2C anyway.

2. Turnover rates of macromolecular sub-complexes:

"Proteins within a complex are vastly ($P < 10^{-99}$, Wilcoxon-Mann-Whitney test) closer in Euclidian space than the rest of the proteins (Fig. 3a)."

Is this only apparent for the combination of synthesis and decay rates or could it also be explained by one of the opposing mechanisms. If not, which of the rates has a stronger influence? This must be addressed and at least discussed as the authors otherwise state that synthesis plays the predominant role.

The reviewer is correct that the synthesis and degradation rate could, in principle, have different contributions as understood from figure 3A. However, since this was a confirmatory point about proteins participating in macromolecular complexes, and reviewer 1 suggested to leave out some confirmatory points, we have taken this out.

3. Temporal correlation between the transcriptome and the proteome:

This referee has concerns about the figures: please label all figures in order to clarify the results. E.g. add legends, add information to what is correlated with what for the correlation plots. Generally, for correlation and box plots, please indicate the number of data points for error assessments.

We agree with the reviewer and have labeled the box-plots, added figure legend and added the number of data points in figure legend.

4. Figure 4A: labeling of axis and description is not clear. Of how many data points were the box plots made of? Please better show scatterplots of averaged replicates.

We have included the number of data points in the figure legend, however we do not agree that it would better to show scatterplots, since then the standard deviation between replicates is then obscured.

5. "When we examined the correlation of mRNA and proteins expression change at each of the five time points, we observed that the relationship nearly reached a steady state after 24h differentiation, suggesting that extensive post transcriptional regulation is taking place during early differentiation (Fig. 4b)."

If I understood this correctly the mRNA and protein levels compared are each measured at the same time points. In a dynamic adaptation to a certain stimulus, you would however in any case expect a lag between changed mRNA levels and their translation if the steady state is not reached. How can the authors conclude extensive post-transcriptional regulation? This should be deleted or clarified with solid arguments.

It is correctly understood that we compared the mRNA and protein expression changes to each other at 6, 12, 24 and 48 h. We agree with this referee that the lag between mRNA and protein expression could also be a viable explanation for why a steady state is first reached after 24 h differentiation. We have therefore changed the sentence to:

p. 10 When we examined the correlation of mRNA and proteins expression change at each of the five time points, we observed that the relationship nearly reached a steady state after 24h differentiation, suggesting a lag between mRNA protein expression changes and/or that extensive post transcriptional regulation is taking place during early differentiation (Fig. 4b).

6. "with less than 6 % of the genes showing opposite regulation at 48 hours differentiation" - this must be clarified.

We agree with the reviewer this was not clear and have changed the sentence to:

p.10 Interestingly, however, proteins and mRNAs that were being up- or down-regulated during differentiation were much more highly correlated, with less than 6 % of the significantly changing genes showing anti-correlation at 48 h differentiation, suggesting that post-transcriptional regulation such as miRNA and 3' and 5' UTRs is mainly just fine tuning the levels of these proteins after 48 h stimulation.

7. Modeling the control of protein expression:

"If one looks first at the effect of synthesis and degradation rates on overall protein expression, it is

obvious that if the synthesis and degradation rates of a protein are equal then the expression of the protein is stable (Fig. 5a-b)."

If I understood this correctly the word stable is used incorrectly in this context. I think the authors mean "steady state levels". Please clarify.

Thank you for pointing this out. We have now changed the text to use 'steady state'

8. Fig. 5a/b: What are the conclusions from these panels? Are these data anti-correlated? If yes this is a nice finding because it shows that cells synthesize proteins they need for differentiation and at the same time stabilize them. Proteins that are not needed are not synthesized any more and also show higher degradation rates. If the authors agree I recommend emphasizing this.

We agree with the reviewer it was not completely clear what the conclusions from these were, therefore we have added the following sentence:

p. 11 If one looks first at the effect of synthesis and degradation rates on overall protein expression, it is obvious that if the synthesis and degradation rates of a protein are equal then the expression of the protein is at a steady state, whereas the net protein expression change is the result of the relationship between the synthesis and degradation rates (Fig. 5a-b).

However we would like to emphasize that one should NOT conclude from this figure anything about protein synthesis or degradation changes in response to differentiation, since it only shows the relative synthesis and degradation rates during differentiation.

9. Fig. 5d is not clear to me. I do not see an increase in the predictive power of synthesis or degradation rates and I cannot see which part of the text belongs to which part in the figure. This must be clarified or taken out.

We had tried to make the point that when trying to predict the ultimate changes in protein expression protein expression from mRNA changes, one gains more predictive power if one also knows the relative synthesis and degradation rates of the proteins, which can be seen in the bar-plot as the whole bar being much larger than just the bars representing mRNA. In an attempt to clarify this we have changed the order of the parameters on the bar chart, so that the contribution to the model of mRNA is at the bottom with contributions from synthesis and degradation rates above, so the increase in prediction powers can easily be seen as the increase in height of the bars above the mRNA.

References

- Forrest ARR, Kanamori-Katayama M, Tomaru Y, Lassmann T, Ninomiya N, Takahashi Y, de Hoon MJL, Kubosaki A, Kaiho A, Suzuki M, Yasuda J, Kawai J, Hayashizaki Y, Hume DA & Suzuki H (2009) Induction of microRNAs, mir-155, mir-222, mir-424 and mir-503, promotes monocytic differentiation through combinatorial regulation. *Leukemia* **24**: 460–466
- Lee MV, Topper SE, Hubler SL, Hose J, Wenger CD, Coon JJ & Gasch AP (2011) A dynamic model of proteome changes reveals new roles for transcript alteration in yeast. *Molecular Systems Biology* **7**: 1–12
- Prieto J, Eklund A & Patarroyo M (1994) Regulated expression of integrins and other adhesion molecules during differentiation of monocytes into macrophages. *Cell. Immunol.* **156**: 191–211
- Suzuki H, Forrest ARR, van Nimwegen E, Daub CO, Balwiercz PJ, Irvine KM, Lassmann T, Ravasi T, Hasegawa Y, de Hoon MJL, Katayama S, Schroder K, Carninci P, Tomaru Y, Kanamori-Katayama M, Kubosaki A, Akalin A, Ando Y, Arner E, Asada M, et al (2009) The transcriptional network that controls growth arrest and differentiation in a human myeloid leukemia cell line. *Nat. Genet.* **41**: 553–562
- Vizcaíno JA, Côté RG, Csordas A, Dienes JA, Fabregat A, Foster JM, Griss J, Alpi E, Birim M, Contell J, O'Kelly G, Schoenegger A, Ovelleiro D, Pérez-Riverol Y, Reisinger F, Ríos D,

Wang R & Hermjakob H (2013) The PRoteomics IDentifications (PRIDE) database and associated tools: status in 2013. *Nucleic Acids Research* **41**: D1063–9

2nd Editorial Decision

19 August 2013

Thank you again for submitting your work to Molecular Systems Biology. We have now heard back from the two referees who accepted to evaluate your revised manuscript. As you will see, the referees are now satisfied with the modifications made and support publication of the work. As a matter of course, please make sure that you have correctly followed the instructions for authors as given on the submission website.

REFeree REPORTS

Reviewer #1

The authors have answered my main concerns in a substantially revised version of the manuscript, and have deposited their data to a public repository. I believe that the revised manuscript makes a contribution worthy of publication.

As a minor side note, I would encourage them to pay attention to the visual aspects of the figures when uploading their final version.

Reviewer #3

The authors made an effort to address my concerns. I recommend publication. I apologize that one of my comments was based on a misunderstanding but I hope this helped the authors to more clearly present the data.